# LOW-RANK MATRIX RECOVERY WITH UNKNOWN CORRESPONDENCE

## ABSTRACT

We study a matrix recovery problem with unknown correspondence: given the observation matrix $M_o = [A, \tilde{P}B]$, where $\tilde{P}$ is an unknown permutation matrix, we aim to recover the underlying matrix $M = [A, B]$. Such problem commonly arises in many applications where heterogeneous data are utilized and the correspondence among them are unknown, e.g., due to privacy concerns. We show that it is possible to recover $M$ via solving a nuclear norm minimization problem under a proper low-rank condition on $M$, with provable non-asymptotic error bound for the recovery of $M$. We propose an algorithm, $M^3O$ (Matrix recovery via Min-Max Optimization) which recasts this combinatorial problem as a continuous minimax optimization problem and solves it by proximal gradient with a Max-Oracle. $M^3O$ can also be applied to a more general scenario where we have missing entries in $M_o$ and multiple groups of data with distinct unknown correspondence. Experiments on simulated data, the MovieLens 100K dataset and Yale B database show that $M^3O$ achieves state-of-the-art performance over several baselines and can recover the ground-truth correspondence with high accuracy.

## 1 INTRODUCTION

In the era of big data, one usually needs to utilize data gathered from multiple disparate platforms when accomplishing a specific task. However, the correspondence among the data samples from these different sources are often unknown due to either missing identity information or privacy reasons (Unnikrishnan et al., 2018; Gruteser et al., 2003; Das & Lee, 2018). Examples include the multi-image matching problem studied in (Ji et al., 2014; Zeng et al., 2012; Zhou et al., 2015), the record linkage problem (Chan & Loh, 2001) and the federated recommender system (Yang et al., 2020).

In the simplest scenario, we have two data matrices $A = [a_1, ..., a_n]^\top$, $B = [b_1, ..., b_n]^\top$ with $a_i \in \mathbb{R}^{m_A}$ and $b_i \in \mathbb{R}^{m_B}$, which are from two different platforms (data sources). As discussed above, the correspondence $(a_i, b_i)$ may not be available, and thereby the goal is to recover the underlying correspondence between $a_1, ..., a_n$ and $b_{\tilde{\pi}(1)}, ..., b_{\tilde{\pi}(n)}$, where $\tilde{\pi}(\cdot)$ denotes an unknown permutation. We can translate such problem described above as a matrix recovery problem, i.e., to recover the matrix $M = [A, B]$ from the permuted observation $M_o = [A, \tilde{P}B]$, where $\tilde{P} \in \mathcal{P}_n$ is an unknown permutation matrix and $\mathcal{P}_n$ denotes the set of all $n \times n$ permutation matrices. We term this problem as **M**atrix **R**ecovery with **U**nknown **C**orrespondence (**MRUC**).

Inspired by the classical low-rank model for matrix recovery (Wright & Ma, 2021; Mazumder et al., 2010; Hastie et al., 2015), we especially focus on the scenario where the matrix $M$ features a certain low-rank structure. Such low-rank model has achieved great success in many applications like the recommender system (Schafer et al., 2007; Mazumder et al., 2010) and the image recovery and alignment problem (Zeng et al., 2012; Zhou et al., 2015). By denoting $B_o = \tilde{P}B$, we want to solve the following rank minimization problem for MRUC,

$$\min_{P \in \mathcal{P}_n} \text{rank}([A, PB_o]). \tag{1}$$

**Practical applications.** It is known that the recommender system often suffers from data sparsity (Zhang et al., 2012) because users typically only provide ratings for very few items. To enlarge the set of observable ratings for each user, we may harness extra data from multiple platforms (Netflix, Amazon, Youtube, etc.). One classical work on this problem is the multi-domain recommender system considered in (Zhang et al., 2012). Unfortunately, their work neglects a crucial issue that

data from these diverse platforms (or domains) are not always well aligned for two primary reasons. The first is that the same user may use different identities, or even leave nothing about their identities, on these platforms. Another reason is that, those platforms are not allowed to share with each other the identity information about their users for preserving privacy. Another application is the visual permutation learning problem (Santa Cruz et al., 2017), where one needs to recover the original image from the *shuffled* pixels. Both of the two applications give rise to a challenging extension of the MRUC problem, where we not only need to recover multiple correspondence across different data sources, but also face the difficulty of dealing with the missing values in data matrix .

**Relationship to the multivariate unlabeled sensing problem.** Problem (1) is closely related to the **M**ultivariate **U**nlabeled **S**ensing (**MUS**) problem, which has been studied in (Pananjady et al., 2017; Zhang et al., 2019a;b; Zhang & Li, 2020; Slawski et al., 2020b;a). Specifically, the MUS is the multivariate linear *regression* problem with unknown correspondence, i.e., it solves

$$\min_{P \in \mathcal{P}_n, W \in \mathbb{R}^{m_2 \times m_1}} \|Y - PXW\|_F^2, \tag{2}$$

where $W \in \mathbb{R}^{m_2 \times m_1}$ is the regression coefficient matrix, $Y \in \mathbb{R}^{n \times m_1}$ and $X \in \mathbb{R}^{n \times m_2}$ denotes the output and the permuted input respectively, and $\|\cdot\|_F$ is the matrix Frobenius norm. In fact, a concurrent work (Yao et al., 2021) studies the same rank minimization problem as (1), but their approach is to solve it using the algorithm developed for MUS problem. Despite of the similarity to the MUS problem, we remark that MRUC problem has it own distinct features and, as shown in Section 4, the algorithm for the MUS algorithm can not be directly and effectively applied, especially when there are multiple unknown correspondence and missing entries to be considered.

**Related works.** To the best of our knowledge, the concurrent and independent (Yao et al., 2021) is the only work that also considers the MRUC problem. Theoretically, (Yao et al., 2021) showed that there exists an non-empty open subset $U \subseteq \mathbb{R}^{n \times (m_1+m_2)}$, such that $\forall M \in U$, solving (1) is bound to recover the original correspondence. However, such results only prove its existence for the subset $U$ and do not provide a concrete characterization. Regarding the algorithm design, (Yao et al., 2021) follows the idea of (Slawski et al., 2020b;a) and treats problem (1) heuristically as a MUS problem. However, there are two main drawbacks in their algorithm that largely limit its practical value. First, in the dense permutation scenario, it ignores the interaction among the shuffled columns and hence can not utilize the prior knowledge on the unknown permutation to have an improved performance; Second, their method can not deal with data with missing values.

**Contributions of this work.** Our contributions in this work lie in both theoretical and practical aspects. Theoretically, we are the first to rigorously study how the rank of the data matrix is perturbed by the permutation, and show that problem (1) can be used to recover a generic low-rank random matrix almost surely. Besides, we also propose a nuclear norm minimization problem as a surrogate for problem (1). The most important theoretical result in this work is that we provide a non-asymptotic analysis to bound the error of the nuclear norm minimization problem under a mild assumption. Practically, we propose an efficient algorithm M³O that solves the nuclear norm minimization problem, which overcomes the aforementioned two shortcomings in (Yao et al., 2021). Notably, M³O works very well even for an extremely difficult task, where we need to recover multiple unknown correspondence from the data that are densely permuted and contain missing values. We remark that this is so far a challenging problem unexplored in the existing literature.

**Outline.** For conciseness, we will first study the MRUC problem with single unknown correspondence, and then show that the theoretical results and the algorithm can be readily extended to the more complicated scenarios. We start with building the theoretical results for (1) and its convex relaxation in Section 2. Then, the algorithm is developed in Section 3. The simulation results are presented in Section 4 and the conclusions are drawn in Section 5.

**Notations.** Given two matrices $X, Y \in \mathbb{R}^{n \times m}$, we denote $\langle X, Y \rangle = \sum_{i=1}^{n} \sum_{j=1}^{m} X_{ij} Y_{ij}$ as the matrix inner product. We denote $X(i)$ as the $i$th row of the matrix $X$ and $X(i,j)$ as the element at the $i$th row and the $j$th column. We denote $\mathbf{1}_m \in \mathbb{R}^m$ and $\mathbf{1}_{n \times m} \in \mathbb{R}^{n \times m}$ as the all-one vector and matrix, respectively, and $I_n$ be the $n \times n$ identity matrix. For $\alpha \in \mathbb{R}^m$, $\beta \in \mathbb{R}^n$, we define the operator $\oplus$ as $\alpha \oplus \beta = \alpha \mathbf{1}_n^\top + \mathbf{1}_m \beta^\top \in \mathbb{R}^{m \times n}$. We denote $\| \cdot \|_*$ as the nuclear norm for matrices. For vectors, we denote $\| \cdot \|_0, \| \cdot \|_1$ as the zero norm and 1-norm respectively.

## 2 MATRIX RECOVERY VIA A LOW-RANK MODEL

**How is the matrix rank perturbed by the row permutation?** To answer this fundamental question, we first introduce the cycle decomposition of a permutation.

**Definition 1** (Cycle decomposition of a permutation (Dummit & Foote, 1991)). *Let $S$ be a finite set and $\pi(\cdot)$ be a permutation on $S$. A cycle $(a_1, ..., a_n)$ is a permutation sending $a_j$ to $a_{j+1}$ for $1 \leq j \leq n-1$ and $a_n$ to $a_1$. Then a cycle decomposition of $\pi(\cdot)$ is an expression of $\pi(\cdot)$ as a union of several disjoint cycles[1].*

It can be verified that any permutation on a finite set has a unique cycle decomposition (Dummit & Foote, 1991). Therefore, we can define the *cycle number* of a permutation $\pi(\cdot)$ as the number of disjoint cycles with length greater than 1, which is denoted as $C(\pi)$. We also define the non-sparsity of a permutation as the Hamming distance between it and the original sequence, i.e., $H(\pi) = \sum_{s \in S} \mathbb{I}[\pi(s) \neq s]$. It is obvious that $H(\pi) > C(\pi)$ if $\pi$ is not an identity permutation. As a simple example, we consider the permutation $\pi(\cdot)$ that maps the sequence (1,2,3,4,5,6) to (3,1,2,5,4,6). Now the cycle decomposition for it is $\pi(\cdot) = (132)(45)(6)$, and $C(\pi) = 2$, $H(\pi) = 5$.

In all the following theoretical results, we denote the original matrix as $M = [A, B] \in \mathbb{R}^{n \times m}$ with $A \in \mathbb{R}^{n \times m_A}$, $B \in \mathbb{R}^{n \times m_B}$, and $\text{rank}(M) = r$, $\text{rank}(A) = r_A$, $\text{rank}(B) = r_B$. We denote the corresponding permutation as $\pi_P(\cdot)$ for any permutation matrix $P \in \mathcal{P}_n$. The following proposition says that the perturbation effect of a permutation $\pi$ on the rank of $M$ becomes stronger, if $\pi$ permutes more rows and contains less cycles.

**Proposition 1.** $\forall P \in \mathcal{P}_n$, *we have*
$$\text{rank}([A, PB]) \leq \min\{n, m, r_A + r_B, r + H(\pi_P) - C(\pi_P)\}. \tag{3}$$
We have similar result for the case with multiple permutations, which is summarized in Corollary 1 in Appendix A.1. It turns out that, without any assumption on $M$, (3) is the tightest upper bound for the rank of a perturbed matrix. Notably, the following proposition says that the upper bound in (3) is attained with probability 1 for a generic low-rank random matrix.

**Definition 2.** *A probability distribution on $\mathbb{R}$ is called a proper distribution if its density function $p(\cdot)$ is absolutely continuous with respect the Lebesgue measure on $\mathbb{R}$.*

**Proposition 2.** *If the original matrix $M$ is a random matrix with $M = RE$ where $R \in \mathbb{R}^{n \times r}$ and $E \in \mathbb{R}^{r \times m}$ are two random matrices whose entries are i.i.d and follow a proper distribution on $\mathbb{R}$, and $r \leq \min\{\sqrt{\frac{n}{2}}, m_A, m_B\}$, then $\forall P \in \mathcal{P}_n$, the equality*
$$\text{rank}([A, PB]) = \min\{2r, r + H(\pi_P) - C(\pi_P)\} \tag{4}$$
*holds with probability 1.*

**Convex relaxation for the rank function.** Despite the previous theoretical justification for problem (1), it is non-convex and non-smooth. Another crucial issue is that we often have a noisy observation matrix and it is well known that the rank function is extremely sensitive to the additive noise. In this paper, we assume that the observation matrix is corrupted by i.i.d Gaussian additive noise, i.e.,
$$M_o = [A_o, B_o] = [A, \tilde{P}B] + W, \text{ where } W(i, j) \sim \mathcal{N}(0, \sigma^2),$$
where $\sigma^2$ reflects the strength of the noise. We first denote the singular values of a matrix $X \in \mathbb{R}^{n \times m}$ as $\sigma_X^1, ..., \sigma_X^k$ where $k = \min\{n, m\}$. Since $\text{rank}(X) = \|[\sigma_X^1, ..., \sigma_X^k]\|_0$, from Proposition 2 we can view the perturbation effect of a permutation to a low-rank matrix as breaking the sparsity of its singular values. This view leads naturally to the well-known 1-norm minimization problem which has been proven robust to additive noise and can yield a sparse solution (Wright & Ma, 2021), i.e.,
$$\min_{P \in \mathcal{P}_n} \|[A_o, PB_o]\|_* = \|[\sigma_{M_o}^1, ..., \sigma_{M_o}^k]\|_1. \tag{5}$$

Since for an arbitrary matrix, the 1-norm of its singular values is equivalent to its nuclear norm, we refer problem (5) as the nuclear norm minimization problem.

**Theoretical justification for the nuclear norm.** Nuclear norm has a long history used as a convex surrogate for the rank, and it has been theoretically justified for applications like low-rank matrix completion (Candès & Tao, 2010; Wright & Ma, 2021). It is also important to see whether the nuclear norm is still a good surrogate for the rank minimization problem (1). In this work, we establish a sufficient condition on $A$ and $B$ under which problem (5) is provably justified for correspondence recovery. We denote $A = \sum_{i=1}^{r_A} \sigma_A^i u_A^i v_A^{i\top}$, $B = \sum_{i=1}^{r_B} \sigma_B^i u_B^i v_B^{i\top}$ as the singular values decomposition of $A$ and $B$, where the $\sigma_A^i$ and $\sigma_B^i$ are the non-zero singular values.

Firstly, from the definition of nuclear norm, it can be simply verified for any $P \in \mathcal{P}_n$ that
$$-Z/N \leq (\|[A, PB]\|_* - \|M\|_*)/\|M\|_* \leq Z/N, \tag{6}$$
where we denote $N = \max\{\|A\|_*, \|B\|_*\}$ and $Z = \min\{\|A\|_*, \|B\|_*\}$. The inequality (6) indicates that $A$ and $B$ should have comparable magnitude, i.e., $\|A\|_* \approx \|B\|_*$, otherwise the influence of the permutation will be less significant. Therefore, we are interested in the scenario where the singular values of $A$ and $B$ are comparable, which is described as the following Assumption 1.

---

[1] Two cycles are disjoint if they do not have common elements

**Assumption 1.** *There exists a constant $\epsilon_1 \geq 0$ such that*

$$|\sigma_A^i - \sigma_B^i| \leq \epsilon_1, \ \forall i = 1, ..., r, \tag{7}$$

*where we denote that $\sigma_A^i = 0$ if $i > r_A$, and $\sigma_B^i = 0$ if $i > r_B$.*

Similar to the matrix rank, we also need a proper low-rank assumption on the matrix $M$ for the nuclear norm. In this work, we particularly study the scenario that the left singular vectors of $A$ and $B$ are similar, which we formally describe as Assumption 2. We refer Assumption 2 as a proper low-rank assumption, because it indicates that the column space of $M$ can be approximated by the column space of one of its submatrices.

**Assumption 2.** *There exists a constant $\epsilon_2 \geq 0$ such that*

$$\|u_A^i - u_B^i\| \leq \epsilon_2, \forall i = 1, ..., T, \tag{8}$$

*where we denote $T = \min\{r_A, r_B\}$.*

Furthermore, we also need that all the column singular vectors $u_A^1, ..., u_A^T, u_B^1, ..., u_B^T$ are variant under any $P \in \mathcal{P}_n$ with $P \neq I_n$: we define a vector $u \in \mathbb{R}^n$ to be variant under a $P \in \mathcal{P}_n$ if $Pu \neq u$. One simple and weak condition for a vector $u$ to satisfy such property is that $u$ dose not contains duplicated elements, which leads to the following Assumption 3.

**Assumption 3.** *There exists a constant $\epsilon_3 \geq 0$ such that*

$$\min_{u \in U} \min_{i \neq j} |u(i) - u(j)| \geq \epsilon_3 > 0, \tag{9}$$

*where $U = \{u_A^1, ..., u_A^T, u_B^1, ..., u_B^T\}$.*

In summary, the assumptions mentioned above feature a typical low-rank structure in $M$, and implies that the nuclear norm of $M$ is sensitive to permutation. With the three assumptions, we have the following important theorem, which provides high probability bound for the approximation error of (5).
We denote the solution to (5) as $P^*$, and let $\pi^*$ and $\tilde{\pi}$ be the corresponding permutation to the permutation matrices $P^{*\top}$ and $\tilde{P}$, respectively. We define the difference between the two permutations $\pi^*$ and $\tilde{\pi}$ as the *Hamming* distance

$$d_H(\pi^*, \tilde{\pi}) \stackrel{\text{def.}}{=} \sum_{i=1}^n \mathbb{I}(\pi^*(i) \neq \tilde{\pi}(i)).$$

**Theorem 1.** *Under Assumptions 1, 2 and 3, if additionally $\epsilon_1 \leq \frac{M}{4r}$, $\epsilon_2 \leq \min\{\frac{1}{2\sqrt{2T}}, \frac{\sqrt{2}M}{2N}\}$, and $\sigma \leq \frac{M}{16L^2}$, then the following bound for the Hamming distance*

$$d_H(\pi^*, \tilde{\pi}) \leq \frac{2}{\epsilon_3^2} \left( 2 - \left( \frac{\sqrt{2}D}{D + (\sqrt{2}+2)\epsilon_1 r + \sqrt{2}\epsilon_2 N + 2\sqrt{2DL\sigma}} - \sqrt{T}\epsilon_2 \right)^2 \right) \tag{10}$$

*holds with probability at least $1 - 2\exp\{-\frac{D}{8L\sigma}\}$, where $L = \max\{n, m\}$, $D = \|A\|_* + \|B\|_*$.*

The proof to all the aforementioned theoretical results are provided in Appendix A.1.

**Remark 1.** From Theorem 1 we can see that when $\epsilon_3 > 0$, and $\epsilon_1 \to 0$, $\epsilon_2 \to 0$, $\sigma \to 0$, the error $d_H(\pi^*, \tilde{\pi})$ will converge to zero with probability 1. Furthermore, we can also discover that the correspondence can be difficult to recover when:

- The rank of original matrix $M$ is high, which can be seen from (10).
- The magnitude of $A$ and $B$ w.r.t rank or nuclear norm are not comparable, which can be seen from (6) and (7).
- The strength of noise is high, which can be seen from (10) and the probability in Theorem 1.

Notably, the numerical experiments in Section 4.1 corroborate our claims as well.

**Remark 2.** Additionally, from the proof of Theorem 1 we find that the fundamental reason for the success of (5) is that if $M$ satisfies the previous assumptions, we have

$$\|[A, PB]\|_* / \|M\|_* \approx O\left((1 - H(\pi_P)/2n)^{-\frac{1}{2}}\right). \qquad (11)$$

In many applications, we can only observe part of the full data. Therefore, it is also worthwhile to investigate whether (11) still holds when we can only access a small subset of the entries in $M_o$. Notably, Figure 1 gives the positive answer and shows that the relationship (11) is gracefully degraded when the percentage of observable entries is decreasing. This phenomenon is remarkable since it indicates the original correspondence can be recovered

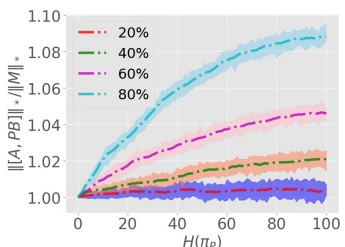

Figure 1: The relationship (11) under different percentages of observable entries.

from only part of the full data. The matrices used to generate Figure 1 are the same as those in Section 4.1, and the nuclear norm is computed approximately by first filling the missing entries using Soft-Impute algorithm (Mazumder et al., 2010).

## 3 ALGORITHM

In this section, we consider the scenario with missing values, i.e., our observed data is $\mathcal{P}_\Omega(M_o) = \mathcal{P}_\Omega([A_o, B_o])$, where $\mathcal{P}_\Omega$ is an operator that selects entries that are in the set of observable indices $\Omega$. In this scenario, problem (5) can not be directly used since the evaluation of the nuclear norm and optimization of the permutation are coupled together. Inspired by the matrix completion method (Hastie et al., 2015; Mazumder et al., 2010), we propose to solve an alternative form of (5) as follows,

$$\min_{\widehat{M} \in \mathbb{R}^{n \times m}} \min_{P \in \mathcal{P}_n} \left\| \mathcal{P}_\Omega([A_o, PB_o]) - \mathcal{P}_\Omega(\widehat{M}) \right\|_F^2 + \lambda \left\| \widehat{M} \right\|_*, \qquad (12)$$

where $\lambda > 0$ is the penalty coefficient. We denote that $\widehat{M} = [\widehat{M}_A, \widehat{M}_B]$ and $\widehat{M}_A, \widehat{M}_B$ are the two submatrices with the same dimension as $A_o$ and $B_o$ respectively. We can write (12) equivalently as

$$\min_{\widehat{M} \in \mathbb{R}^{n \times m}} \min_{P \in \mathcal{P}_n} \left\| \mathcal{P}_\Omega(A_o) - \mathcal{P}_\Omega(\widehat{M}_A) \right\|_F^2 + \langle C(\widehat{M}_B), P \rangle + \lambda \left\| \widehat{M} \right\|_*, \qquad (13)$$

where $C(\widehat{M}_B) \in \mathbb{R}^{n \times n}$ is the pairing cost matrix with

$$C(\widehat{M}_B)(i,j) = \sum_{(j, j'') \in \Omega} \left( \widehat{M}_B(i, j'') - B_o(j, j'') \right)^2, \forall i, j = 1, ..., n.$$

**Baseline algorithm.** A conventional strategy to handle an optimization problem like (13) is the alternating minimization or the block coordinate descent algorithm (Abid et al., 2017). Specifically, it executes the following two updates iteratively until it converges.

$$\widehat{M}^{\text{new}} \leftarrow \underset{\widehat{M} \in \mathbb{R}^{n \times m}}{\arg\min} \left\| \mathcal{P}_\Omega([A_o, \widehat{P}^{\text{old}} B_o]) - \mathcal{P}_\Omega(\widehat{M}) \right\|_F^2 + \lambda \left\| \widehat{M} \right\|_*, \qquad (14)$$

$$\widehat{P}^{\text{new}} \leftarrow \underset{P \in \mathcal{P}_n}{\arg\min} \langle C(\widehat{M}_B^{\text{new}}), P \rangle. \qquad (15)$$

The first update step (14) is a convex optimization problem and can be solved by the proximal gradient algorithm (Mazumder et al., 2010). The second update step (15) is actually a discrete optimal transport problem which can be solved by the classical Hungarian algorithm with time complexity $O(n^3)$ (Jonker & Volgenant, 1986). However, as we will see in the Section 4, this algorithm performs poorly, and it is likely to fall into an undesirable local solution quickly in practice. Specifically, the main reason is that the solution of (15) is often not unique and a small change in $\widehat{M}_B$ would lead to large change of $\widehat{P}$. To address this issue, we propose a novel and efficient algorithm M³O algorithm based on the entropic optimal transport (Peyré et al., 2019) and min-max optimization (Jin et al., 2020).

**Smoothing the permutation with entropy regularization.** For any $a \in \mathbb{R}^n, b \in \mathbb{R}^m$, we define

$$\Pi(a, b) = \{ S \in \mathbb{R}^{n \times m} : S\mathbf{1}_m = a, S^\top \mathbf{1}_n = b, S(i,j) \geq 0, \forall i, j \},$$

which is also known as the Birkhoff polytope. The famous Birkhoff-von Neumann theorem (Birkhoff, 1946) states that the set of extremal points of $\Pi(\mathbf{1}_n, \mathbf{1}_n)$ is equal to $\mathcal{P}_n$. Inspired by (Xie et al., 2021) and the interior point method for linear programming (Bertsekas, 1997), in order to smooth the optimization process of the baseline algorithm, we relax $P$ from being an exact permutation matrix, i.e., to keep $P$ staying inside the Birkhoff polytope $\Pi(\mathbf{1}_n, \mathbf{1}_n)$. That is, we propose to replace the combinatorial problem (15) with the following continuous optimization problem

$$\min_{P \in \Pi(\mathbf{1}_n, \mathbf{1}_n)} \langle C(\widehat{M}_B), P \rangle + \epsilon H(P), \qquad (16)$$

where $H(P) \stackrel{\text{def.}}{=} \sum_{i,j} P(i,j)(\log(P(i,j)) - 1)$ is the matrix negative entropy and $\epsilon > 0$ is the regularization coefficient. Notably, (16) is also known as the Entropic Optimal Transport (EOT) problem. According to (Peyré et al., 2019), (16) is a strongly convex optimization problem with respect to 1-norm and can be solved roughly in the $O(n^2)$ complexity per iteration by the Sinkhorn algorithm. Specifically, the Sinkhorn algorithm solves the dual problem of (16),

$$\max_{\alpha, \beta \in \mathbb{R}^n} W_\epsilon(\widehat{M}_B, \alpha, \beta) \stackrel{\text{def.}}{=} \langle \mathbf{1}_n, \alpha \rangle + \langle \mathbf{1}_n, \beta \rangle - \epsilon \left\langle \mathbf{1}_{n \times n}, \exp\left\{ \frac{\alpha \oplus \beta - C(\widehat{M}_B)}{\epsilon} \right\} \right\rangle, \quad (17)$$

which reduces the variables dimension from $n^2$ to $2n$ and is thus greatly favorable in the high dimension scenario. By substituting the inner minimization problem of (13) with (16), we end up with solving the following unconstrained min-max optimization problem

$$\min_{\widehat{M}} \max_{\alpha, \beta} \left\| A - \widehat{M}_A \right\|_F^2 + W_\epsilon(\widehat{M}_B, \alpha, \beta) + \lambda \left\| \widehat{M} \right\|_*. \quad (18)$$

Follows the idea of (Jin et al., 2020), we consider to adopt a proximal gradient algorithm with a Max-Oracle for (18). Specifically, we employ the Skinhorn algorithm (Peyré et al., 2019) as the Max-Oracle to retrieve an $\varepsilon$-good solution of the inner max problem (17). We summarize our proposed algorithm M$^3$O (**M**atrix recovery via **M**in-**M**ax **O**ptimization) in Algorithm 1, where $\text{prox}_{\lambda \|\cdot\|_*}(\cdot)$ is the proximal operator of nuclear norm and $\rho_k$ is the gradient stepsize. The convergence property of M$^3$O can be obtained by following (Jin et al., 2020), which shows that, with a decaying stepsize, M$^3$O is bound to converge to an $\varepsilon$-good Nash equilibrium within $O(\varepsilon^{-2})$ iterations.

---

**Algorithm 1:** M$^3$O

---

**1** **while** not converged **do**

**2**     For the tolerance $\varepsilon$, run the Sinkhorn algorithm to find $\alpha^*$, $\beta^*$ such that

$$W_\epsilon(\widehat{M}_B^k, \alpha^*, \beta^*) > \max_{\alpha, \beta} W_\epsilon(\widehat{M}_B^k, \alpha, \beta) - \varepsilon;$$

**3**     Perform $\widehat{M}^{k+1} \leftarrow \text{prox}_{\lambda \|\cdot\|_*}(\widehat{M}^k - \rho_k \nabla_{\widehat{M}} F_\epsilon(\widehat{M}^k, \alpha^*, \beta^*))$, where

$$F_\epsilon(\widehat{M}, \alpha, \beta) \stackrel{\text{def.}}{=} \left\| A - \widehat{M}_A \right\|_F^2 + W_\epsilon(\widehat{M}_B, \alpha, \beta);$$

**4** **end**

---

**Remark 3.** A recent work (Xie et al., 2020) proposes a decaying strategy for the entropy regularization coefficient $\epsilon$ in (16) so that the optimal solutions of (15) and (16) do not deviate too much. Inspired by it, in our practice, we take large $\epsilon$ in the beginning and gradually shrink it by half until the objective function stops improving for $K$ steps.

**Remark 4.** A useful trick is that we should not take large stepsize $\rho_k$ in the early iterations because the permutation matrix could still be far away from the optimal one. However, a small stepsize would lead to slow convergence. Heuristically, we propose an adaptive stepsize strategy that performs well in practice. For the solution of (16) $\widehat{P}_k$ at the $k$th iteration, we compute the two statistics

$$\delta_k = \left\| \widehat{P}_{k-1} - \widehat{P}_k \right\|_F^2 / 2n \text{ and } c_k = \left\| \max_j \widehat{P}_k(\cdot, j) - \mathbf{1}_n \right\|_1 / n.$$

Here $\delta_k$ represents how fast the permutation matrix $\widehat{P}_k$ changes over the iterations, while $c_k$ measures how far the current $\widehat{P}_k$ is close to an exact permutation matrix. Both $\delta_k$ and $c_k$ reflect the confidence on the current found correspondence. Based on them, we set the stepsize as $\rho_{k+1} = (1 - \delta_k)(1 - c_k)^\omega$, where $\omega > 0$ is a tunable parameter which is often set to a value between 0.5 to 3. $\omega$ actually trades off the convergence speed and final performance. The smaller the $\omega$, the faster the convergence. Therefore, a practical way is to start with a small $\omega$, and gradually increase it until the final performance stops improving.

**Remark 5.** As discussed in Section 1, in many cases we have to deal with the problem that involves multiple correspondence, i.e., we need to recover the matrix $M = [A, B_1, ..., B_d]$ from the observation data $\mathcal{P}_\Omega(M_o)$, where

$$M_o = [A_o, B_o^1, ..., B_o^d] = [A, \tilde{P}_1 B_1, ..., \tilde{P}_d B_d] + W,$$

where $\tilde{P}_l \in \mathcal{P}_n$ and $W$ is a noise matrix. We refer such problem as the **d-correspondence** problem. An important observation is that, although the number of possible correspondence increase exponentially as $d$ grows, the complexity of M$^3$O per iteration only linearly increases with $d$ and can be implemented in a fully parallel fashion. Specifically, in this scenario, we solve the problem

$$\min_{\widehat{M}} \min_{P_1,...,P_d} \left\| \mathcal{P}_\Omega(A_o) - \mathcal{P}_\Omega(\widehat{M}_A) \right\|_F^2 + \sum_{l=1}^d \left\{ \langle C(\widehat{M}_{B_l}), P_l \rangle + \epsilon H(P_l) \right\} + \lambda \left\| \widehat{M} \right\|_*, \quad (19)$$

$$\text{s.t. } P_l \in \Pi(\mathbf{1}_n, \mathbf{1}_n), \ l = 1,...,d,$$

where we denote $\widehat{M} = [\widehat{M}_A, \widehat{M}_{B_1}, ..., \widehat{M}_{B_d}]$. Here $\widehat{M}_A$ and $\widehat{M}_{B_l}$ have the same dimension with $A_o$ and $B_o^l$, respectively. One can find that the inner problems for solving $P_l$ are actually decoupled for each $l$, which guarantees an efficient parallel implementation.

**Remark 6.** Since x problem (12) has a similar form to that considered in (Mazumder et al., 2010). We adopt the same tuning strategy of $\lambda$ as in (Mazumder et al., 2010), which suggests that we should start with large $\lambda$ and gradually decrease it.

We relegate more details about M³O to Appendix A.6.

## 4 EXPERIMENTS

In this section, we evaluate our proposed M³O on both synthetic and real-world datasets, including the MovieLens 100K and the Extended Yale B dataset. We also provide an ablation study for the decaying entropy regularization strategy and the adaptive stepsize strategy proposed in Remarks 3 and 4. In all the experiments, we employ the Soft-Impute

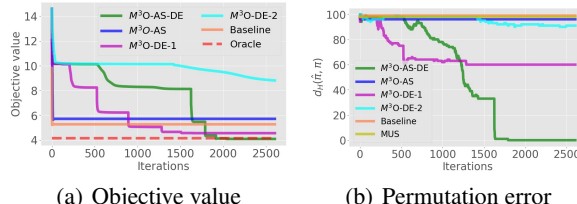

(a) Objective value      (b) Permutation error

Figure 2: Performance of various algorithms on a simulated 1-correspondence problem.

algorithm (Mazumder et al., 2010) as a standard algorithm for matrix completion. Extra experiment details and auxiliary results can be found in Appendix A.9.

**Algorithms.** We denote the following algorithms for comparison in all the experiments:

1. *Oracle*: Running the Soft-Impute algorithm with ground-truth correspondence.

2. *Baseline*: The Baseline algorithm in (14) and (15).

3. *MUS*: Since there is currently no existing algorithm directly applicable to the scenario considered by (19), inspired by (Yao et al., 2021), we modify and extend the algorithm in (Zhang & Li, 2020), which is originally proposed for the MUS problem, to deal with the MRUC problem. The details of the adapted algorithm are provided in Appendix A.8.

### 4.1 SYNTHETIC DATA

We first investigate the property of our proposed M³O algorithm on the synthetic data.

**Data generation.** We generate the original data matrix in this form $M = RE + \eta W$, where $R \in \mathbb{R}^{n \times r}$, $E \in \mathbb{R}^{r \times m}$, $W \in \mathbb{R}^{n \times m}$ and $\eta > 0$ indicates the strength of the additive noise. The entries of $R$, $E$, $W$ are all i.i.d sampled from the $\mathcal{N}(0,1)$. Then we split the data matrix $M$ by $M = [A, B_1, ..., B_d]$ where we denote $A \in \mathbb{R}^{n \times m_A}$, $B_1 \in \mathbb{R}^{n \times m_1}$, ..., $B_d \in \mathbb{R}^{n \times m_d}$ to represent data from $d + 1$ data sources. The permuted observation matrix $M_o$ is obtained by first generating $d$ permutation matrices $P_1, ..., P_d$ randomly and independently, and then computing $M_o = [A, P_1 B_1, ..., P_d B_d]$. Finally, we remove $(1 - |\Omega| \cdot 100\%/(n \cdot m))$ percent of the entries of $M_o$ randomly and uniformly, where $|\Omega|$ indicating the number of observable entries.

**Ablation study.** We denote the following variants of M³O for the ablation study.

1. *M³O-AS-DE*: M³O with both Adpative Stepsize and Decaying Entropy regularization.

2. *M³O-DE*: M³O with Decaying Entropy regularization only. M³O-DE-1 and M³O-DE-2 adopt constant stepsize $\rho_k = 0.5$ and $\rho_k = 0.01$, respectively.

3. *M³O-AS*: M³O with Adpative Stepsize only. The entropy coefficient $\epsilon$ is fixed to 0.0005.

In the following results, we denote $\pi_l$ as the corresponding permutation to $P_l$. We initialize $\widehat{M}$ from Gaussian distribution for the M³O algorithm and its variants. We choose initial $\epsilon$ as 0.1 and $K = 100$ as the default for the decaying entropy regularization, and set $\omega = 3$ as the default for the adaptive stepsize. We also report the achieved objective values of (19) for the tested algorithms, except for the MUS algorithm since it has a different objective. We denote $\hat{\pi}$ as the recovered permutation.

**Results.** Figure 2 displays the result under the setting $\eta = 0.1$, $|\Omega| \cdot 100\%/(n \cdot m) = 80\%$, $n = m = 100$, $r = 5$, $d = 1$, $m_A = 60$ and $m_1 = 40$. The algorithm M³O-AS-DE achieves the best result, and can recover the ground-truth correspondence. M³O-AS behaves similarly to Baseline

and MUS. They all converge to a poor local solution quickly. $M^3$O-DE-1 converges quickly and also falls into a poor local solution due to large stepsize, while $M^3$O-DE-2 adopts a small stepsize and hence suffers from slow convergence. Due to the superiority of $M^3$O-AS-DE over the other variants, in the following results, we refer $M^3$O as $M^3$O-AS-DE for short.

Figure 3 examine $M^3$O on a 1-correspondence problem under different regimes w.r.t $|\Omega|$, $\eta$, $r$ and $m_A/n$. Here we use $m_A/n$ to control the difference of the magnitude of the submatrices. As we can see, the results are well aligned with our prediction in Remarks 1 and 2.

Finally, we examine $M^3$O on a few d-correspondence problems. See Table 1 for various results, where we set $r = 5$ and $\varepsilon = 0.1$. Notice that for the 4-correspondence problem in the table, there are $(100!)^4$ possible correspondence. Even for such a difficult problem, $M^3$O is able to recover 61.5% of the ground-truth correspondence with a good initialization.

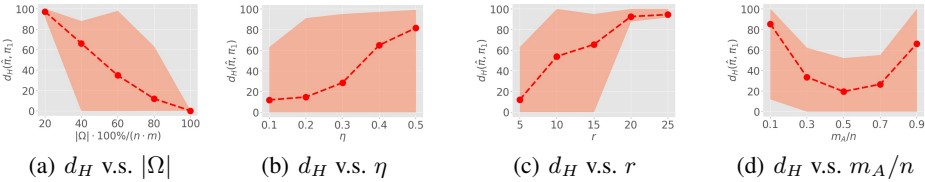

| (a) $d_H$ v.s. $|\Omega|$ | (b) $d_H$ v.s. $\eta$ | (c) $d_H$ v.s. $r$ | (d) $d_H$ v.s. $m_A/n$ |

Figure 3: Performance of $M^3$O on a 1-correspondence problem under different levels of $|\Omega|$, $\eta$, $r$ and $m_A/n$. The default setting is $|\Omega| \cdot 100\%/(n \cdot m) = 80\%$, $\eta = 0.1$, $n = m = 100$, $r = 5$, $m_A = 60$, and $m_1 = 40$. The mean with minimum and maximum are calculated from 10 different random initializations.

Table 1: Performance of $M^3$O for various d-correspondence problems. The normalized permutation error $\sum_{l=1}^{d} d_H(\hat{\pi}_l, \pi_l)/d$ is reported as mean$\pm$std (min) over 10 different random initializations.

| $(n, m_A, m_1, ..., m_d)$ | $d$ | $|\Omega| \cdot 100\%/(n \cdot m)$ | $\sum_{l=1}^{d} d_H(\hat{\pi}_l, \pi_l)/d$ |
|---|---|---|---|
| (100,40,30,30) | 2 | 40% | $33.35 \pm 32.85$ (0.00) |
| (100,20,40,40) | 2 | 40% | $58.90 \pm 27.21$ (2.00) |
| (100,45,25,25,25) | 3 | 50% | $61.97 \pm 15.41$ (37.33) |
| (100,40,25,25,25,25) | 4 | 60% | $59.90 \pm 13.64$ (38.50) |

## 4.2 MULTI-DOMAIN RECOMMENDER SYSTEM WITHOUT CORRESPONDENCE

In this section, we study the performance of $M^3$O on a real world dataset MovieLens 100K[2], which is a widely used movie recommendation dataset (Harper & Konstan, 2015). In this application, we mainly focus on the metric Root Mean Squared Error (RMSE), i.e.,

$$\text{RMSE} \overset{\text{def.}}{=} \sqrt{\frac{1}{N} \sum_{i,j} (\widehat{M}_{ij} - M_{ij})^2}.$$

**Data.** MovieLens 100K contains 100,000 ratings within the scale 1-5. The ratings are given by 943 users on 1,682 movies. Genre information about movies is also provided. We adopt a similar setting with (Zhang et al., 2012). We extract five most popular genres, which are Comedy, Romance, Drama, Action, Thriller respectively, to define the data from 5 different domains (or platforms). In addition to (Zhang et al., 2012), we randomly permute the indexes of the users from these five domains respectively, so that the correspondence among these data become unknown. In this way, the problem belongs to the 4-correspondence problem as discussed before. The ratings are split randomly, with 80% of them as the training data and the other 20% of them as the test data.

**Algorithms.** We consider the following additional algorithms for comparison.

1. *SIC*: Running the Soft-Impute algorithm independently for the 5 different platforms.

2. *SIR*: Running the Soft-Impute algorithm with Randomly generated correspondence.

**Results.** As discussed in experiments on the simulated data, the exact recovery of correspondence becomes impossible due to the small amount of observable entries. Therefore, in the following experiment, since exact correspondence is not needed, we fix $\epsilon = 0.05$ for $M^3$O. Table 2 shows the results by averaging the RMSE on the test data over 10 different random seeds.

We can first see that the matrix completion with a wrong correspondence, i.e., SIR, can be harmful to the overall performance since it is even worse than the results of SIC. Notably, although the ground-truth correspondence can not be recovered, each platform can still benefit from $M^3$O since it

---

[2]https://grouplens.org/datasets/movielens/100k/

improves the performance over SIC. This is mainly because M$^3$O is still able to correspond similar users for inferring missing ratings. On the contrary, since both Baseline and MUS can only establish an exact one-to-one correspondence for each user, they fail to improve SIC significantly. Remarkably, M$^3$O is only inferior to the Oracle method a little, and even achieves lower test RMSE than the Oracle method on the Comedy genre.

Table 2: Test RMSE of various algorithms on MovieLens 100K

| Method | Comedy | Romance | Drama | Action | Thriller | Total |
|---|---|---|---|---|---|---|
| SIR | 1.0202 | 1.0158 | 0.9808 | 0.9803 | 0.9811 | 0.9944 |
| SIC | 0.9694 | 0.9695 | 0.9317 | 0.9175 | 0.9253 | 0.9418 |
| MUS | 0.9659 | 0.9842 | 0.9423 | 0.9305 | 0.9306 | 0.9485 |
| Baseline | 0.9728 | 0.9562 | 0.9379 | 0.9105 | 0.9145 | 0.9395 |
| M$^3$O | **0.9389** | **0.8787** | **0.9139** | **0.8556** | **0.8567** | **0.8948** |
| Oracle | 0.9444 | 0.7825 | 0.9058 | 0.8176 | 0.8098 | 0.8667 |

### 4.3 VISUAL PERMUTATION RECOVERY

We show that M$^3$O is flexible and can also be used to recover matrix that is not in the form $[A, PB]$. We can see this from the problem formulation in (13), where the cost matrix $C(\cdot)$ can be constructed in other ways as long as it is a function of a permutation. Typically, M$^3$O can be used to solve a challenging face image recovery problem. The original face image with size $180 \times 180$ in Figure 4(a) comes from the Extend Yale B database (Georghiades et al., 2001). The corrupted image is visualized in Figure 4(b), where the pixel blocks with size $30 \times 30$ in the upper left are shuffled randomly, and $30\%$ of the total pixels are removed. This kind of problem is recently considered in (Santa Cruz et al., 2017), which proposes to recover the corrupted image in a data-driven way using convolutional neural networks. However, we show that it is possible to recover the image without additional data by merely exploiting the underlying low-rank structure of the image itself.

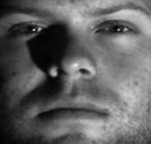 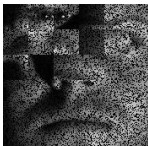

(a) Original      (b) Corrupted

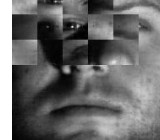 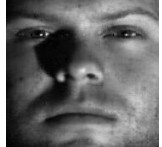

(c) Baseline      (d) M$^3$O

Figure 4: Performance of M$^3$O on a face recovery problem.

This experiment setting is similar to that in (Yao et al., 2021) but the algorithm in (Yao et al., 2021) can not be applied since it can not work with the missing values. The MUS algorithm is also not applicable since this problem can not be written in the form of linear regression problem. From Figure 4(c) and 4(d) we can find that M$^3$O performs better than the Baseline, and can even recover the original orders of pixel blocks. More results similar to the Figure 4 and experiment details are provided in Appendix A.9.

## 5 CONCLUSION

In this paper, we have studied the important MRUC problem where part of the observed submatrix is shuffled. This problem has not been well explored in the existing literature. Theoretically, we are the first to rigorously analyze the role of low-rank model in the MRUC problem, and is also the first to show that minimizing nuclear norm is provably efficient for recovering a typical low-rank matrix. For practical implementations, we propose a highly efficient algorithm, the M$^3$O algorithm, which is shown to consistently achieve the best performance over several baselines in all the tested scenarios.

It is worthwhile to point out that apart from the two applications we have studied in this paper, this problem could arise in more scenarios like the gnome assembly problem (Huang & Madan, 1999), the video pose tracking problem (Ganapathi et al., 2012) and the privacy-aware sensor networks (Gruteser et al., 2003), etc. We believes that our work provides a general framework to deal with unknown correspondence issue in these scenarios.

As we have shown in Figure 3, one major limit of our algorithm is the sensitivity to the initialization. The phenomenon is exacerbated when the additive noise is high or the numbers of observable entries are small. We suggest to try with a few different initialization strategy when applying M$^3$O to a specific task. Finding stable initialization strategy is also an important task for our future works.

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
