# OpenReview forum: "Low-rank Matrix Recovery with Unknown Correspondence"
_ICLR.cc/2022/Conference — ICLR 2022 Submitted_

### Official Review · Reviewer_2Tsx · 2021-10-24

**Correctness:** 3
**Technical Novelty And Significance:** 3
**Empirical Novelty And Significance:** 3
**Recommendation:** 6
**Confidence:** 5

**Main Review:**

Strength:

    1. The problem studied in the paper is interesting and of practical value.
    2. To my knowledge, the theoretical results are novel and interesting, and the proposed algorithm is the first attempt at solving the challenging problem of matrix recovery with missing entries & without correspondences.
    3. The writting is almost clear, except some inaccuracies on the literature review, technical development, and on language usage (pointed below).

Weakness:

    1. The literature review is inaccurate, and connections to prior works are not sufficiently discussed. To be more specific, there are three connections, (i) the connection of (1) to prior works on multivariate unlabeled sensing (MUS), (ii) the connection of (1) to prior works in unlabeled sensing (US), and (iii) the connection of the paper to (Yao et al., 2021).

        (i) In the paper, the authors discussed this connection (i). However, the experiments shown in Figure 2 do not actually use the MUS algorithm of (Zhang & Li, 2020) to solve (1); instead the algorithm is used to solve the missing entries case. This seems to be an unfair comparison as MUS algorithms are not designed to handle missing entries. Did the authors run matrix completion prior to applying the algorithm of (Zhang & Li, 2020)? Also, the algorithm of (Zhang & Li, 2020) is expected to fail in the case of dense permutation.

        (ii) Similar to (i), the methods for unlabeled sensing (US) can also be applied to solve (1), using one column of B_0 at a time. There is an obvious advantage because some of the US methods can handle arbitrary permutations (sparse or dense), and they are immune to initialization. In fact, these methods were used in (Yao et al., 2021) for solving more general versions of (1) where each column of B has undergone arbitrary and usually different permutations; moreover, this can be applied to the d-correspondence problem of the paper. I kindly wish the authors consider incoporating discussions and reviews on those methods.

        (iii) Finally, the review on (Yao et al., 2021) is not very accurate. The framework of (Yao et al., 2021), when applied to (1), means that the subspace that contains the columns of A and B is given (when generating synthetic data the authors assume that A and B come from the same subspace). Thus the first subspace-estimation step in the pipeline of (Yao et al., 2021) is automatically done; the subspace is just the column space of A. As a result, the method of (Yao et al., 2021) can handle the situation where the rows of B are densely shuffled, as discussed above in (ii). Also, (Yao et al., 2021) did not consider only "a single unknown correspondence". In fact, (Yao et al., 2021) does not utilize the prior knowledge that each column of B is permuted by the same permutation (which is the case of (1)), instead it assumes every column of B is arbitrarily shuffled. Thus it is a more general situation of (1) and of the d-correspondence problem. Finally, (Yao et al., 2021)  discusses theoretical aspects of (1) with missing entries, while an algorithm for this is missing until the present work.

    2. In several places the claims of the paper are not very rigorous. For example,
        (i) Problem (15) can be solved via linear assignment algorithms to global optimality, why do the authors claim that "it is likely to fall into an undesirable local solution"? Also I did not find a comparison of the proposed approach with linear assignment algorithms.
        (ii) Problem (16) seems to be "strictly convex", not "strongly convex". Its Hessian has positive eigenvalues everywhere but the minimum eigenvalue is not lower bounded by some positive constant. This is my feeling though, as in the situation of logistic regression, please verify this.
        (iii) The Sinkhorn algorithm seems to use O(n^2) time per iteration, as in (17) there is a term C(hat{M_B}), which needs O(n^2) time to be computed. Experiments show that the algorithm needs > 1000 iterations to converge. Hence, in the regime where n << 1000 the algorithm might take much more time than O(n^2) (this is the regime considered in the experiments). Also I did not see any report on running times. Thus I feel uncomfortable to see the author claim in Section 5 that "we propose a highly efficient algorithm".

    3. Even though an error bound is derived in Theorem 1 for the nuclear norm minimization problem, there is no guarantee of success on the alternating minimization proposal. Moreover, the algorithm requires several parameters to tune, and is sensitive to initialization. As a result, the algorithm has very lage variance, as shown in Figure 3 and Table 1.

Questions:

    1. In (3) the last term r+H(pi_P) and C(pi_P) is very interesting. Could you provide some intuition how it shows up, and in particular give an example?
    2. I find Assumption 1 not very intuitive; and it is unclear to me why "otherwise the influence of the permutation will be less significant". Is it that the unknown permutation is less harmful if the magnitudes of A and B are close?
    3. Solving the nuclear norm minimization program seems to be NP-hard as it involves optimization over permutation matrices and a complicated objective. Is there any hardness result for this problem?

Suggestions: The following experiments might be useful.

    1. Sensitivity to permutation sparsity: As shown in the literature of unlabeled sensing, the alternating minimization of (Abid et al., 2017) works well if the data are sparsely permuted. This might also apply to the proposed alternating minimization algorithm here.
    2. Sensitivity to initialization: One could present the performance as a function of the distance of initialization M^0 to the ground-truth M^*. That is for varying distance c (say from 0.01:0.01:0.1), randomly sample a matrix M^0 so that ||M^0 - M^*||_F < c as initialization, and report the performance accordingly. One would expect that the mean error and variance increases as the quality of initialization decreases.
    3. Sensitivity to other hyper-parameters.

Minor Comments on language usage: (for example)

     1. "we typically considers" in the above of (7)
     2. "two permutation" in the above of Theorem 1
     3. "until converge" in the above of (14)
     4. ......
Please proofread the paper and fix all language problems.

**Summary Of The Paper:**

This paper studies the problem of recovering the ground-truth data matrix from two sub-matrices, where the rows of one submatrix are shuffled by an unknown permutation. For this problem the authors relax the rank minimization formulation into nuclear norm minimization, while it is shown that solving the latter recovers the ground-truth permutation under certain assumptions. The authors further extend the setting to where there could be multiple sub-matrices or missing entries in each sub-matrix, or both, and propose an algorithm of the alternating minization type to solve it. Experiments reveal interesting properties of the problem and algorithm proposed.

**Summary Of The Review:**

I think the paper has made some interesting and important contributions to the problem of matrix recovery with missing entries & without correspondences, while there are still a lot of rooms for improvements. Since this is the first work (to my knowledge) that attempts to solve the very challenging problem of MRUC with missing entries, I would like to encourage its publication, hence, for the moment I recommend for weak accept, with the belief that my concerns will be sufficiently addressed in their revised version.

---

> ### Author Response · Authors · 2021-11-15
> **Thank you very much for all these very constructive comments! We have proofread our paper carefully and modified the review for [Yao, 2021]. We have also added new experiment results into the appendix following your suggestions.**
>
> Thank you very much for all these interesting comments! We have proofread our paper carefully and polished the language.
>
> **Replies to weakness**
>
> 1. (i) As mentioned in Section 4, since there is currently no existing algorithm directly applicable to our problem, the best we can do is to adapt the algorithm  designed for similar problem into our case. In our experiments, we first fill in the missing values using Soft-Impute algorithm and then apply the algorithm in (Zhang & Li 2020). More details can be found in Appendix A.7.
>
>    (ii) We had considered this method before. We thought that the main drawback of this method is that it completely ignores the interaction among the columns of $B$. Although this method can be used to handle an arbitrary number of permutations, it is bound to be inferior to the MUS algorithm in the single permutation scenario since it does not exploit the prior knowledge on the unknown permutation.
>
>    (iii) We agree with the reviewer on most of the statements, and have revised our paper accordingly in the introduction section. However, we politely disagree with the reviewer on one point. We believe that "assume every column of B is arbitrarily shuffled" is not a generalization to the d-correspondence problem. Because, for our d-correspondence problem, the method in [Yao, 2021] developed for the dense permutation scenario can not utilize the prior knowledge on the data matrix that "some of the columns are shuffled by the same permutation" (This prior knowledge is practical, because we know that the data samples from the same platforms are not shuffled.) to have an  improved performance.
>
> 2. (i) We politely point out that there is a misunderstanding. Here we are referring to the optimality of the joint optimization problem (12) instead of (15). This claim is mainly based on the observation that the algorithm in  (14) (15)  often stops whenever an exact permutation matrix is obtained. Such phenomenon is also observed by [Xie et al., 2021].
>
>    (ii) This problem is strongly-convex w.r.t 1-norm on the Birkhoff polytope, and hence can not be checked by the Hessian matrix. We have mentioned this in the revised paper and provided a reference for it.
>
>    (iii) Indeed, the total complexity of our algorithm is $O(n^2T)$ where $T$ is the number of iterations. We regard our algorithm as "highly efficient" for two reasons. Firstly, without the novel entropy relaxation technique, we would have the complexity $O(n^3T)$. Secondly, for the  d-correspondence problem, the number of all possible permutations will increase to $(n!)^d$, whereas our algorithm enjoys a linearly increasing per-iteration complexity O($dn^2$), and is still able to generate  an reasonably near-optimal solution within a polynomial time.
>
> 3. We fully with the reviewers on the two points, and we actually have pointed them out in the paper, see the conclusion section. Addressing these issues is also our most important direction for future works.
>
> **Replies to questions**
>
> 1. We believe that this question can be sufficiently addressed by the proof of Proposition 1. Generally speaking, the idea is to study the null space of $P-I$. As an example, suppose that $C_i$ is one of the cycles in $\pi_P$, and we define a vector $x\in\mathbb{R}^n$ such that $x_j=1$ if $j\in C_i$ else $x_j=0$. Then it is easy to show that $(P-I)x=0$. We can use this property to show that the dimension of range$(P-I)$ is $H(\pi_P)-C(\pi_P)$.
> 2. If  "harmful" means harder to recover, then the answer is  yes. In order to guarantee a successful recovery, we must require that the nuclear norm of $[A,B]$ and $[A,PB]$ are significantly different, so that we can distinguish them. However, as shown in inequality (6), the difference between the nuclear norm of  $[A,B]$ and $[A,PB]$ are bounded by their relative ratio.
> 3. With the continuous relaxation and entropy regularization in Section 3, the problem becomes a convex-concave program and hence has a polynomial time solution. Without these relaxations, the  problem (5) is a convex program over discrete variable. To our knowledge, the most related result is : General convex program over discrete variable is NP-hard in the worst case (such as integer programming).
>
> **Replies to suggestions**
>
> 1. The method in (Abid, 2017) is actually equivalent to the baseline algorithm we propose in Section 3.
> 2. This is a very interesting and useful suggestion! We have conducted this experiment and it indeed worked as the reviewer's prediction. We have also added the experiment result in A.9 of the appendix.
> 3. We have provided more experiments details (discussion for the nuclear norm coefficient) in the appendix for the revised version in A.9 of the appendix.

---

> > ### Comment · Reviewer_2Tsx · 2021-11-23
> > **Thanks for the response**
> >
> > I am not convinced by the rebuttal of the authors. They did not sufficiently address my concerns on literature review, even if I pointed out that. Moreover, they made mistakes in their rebuttal, just because they did not read related papers to a sufficient degree. A live example is that thier review of [Yao, 2021]  led Reviewer CuTB to misunderstanding [Yao, 2021], and the authors showed no efforts in correcting this misunderstanding.
> >
> > The authors wrote in their rebuttal:
> > 1. "Although this method can be used to handle an arbitrary number of permutations, it is bound to be inferior to the MUS algorithm in the single permutation scenario since it does not exploit the prior knowledge on the unknown permutation."
> >
> > 2. "Because, for our d-correspondence problem, the method in [Yao, 2021] developed for the dense permutation scenario can not utilize the prior knowledge on the data matrix that "some of the columns are shuffled by the same permutation" (This prior knowledge is practical, because we know that the data samples from the same platforms are not shuffled.) to have an improved performance."
> >
> > These are incorrect. Please give experiments that support your claim, or give references otherwise. Instead, let me argue my point.
> >
> > As a matter of fact, without utilizing the piror knowledge that "some of the columns are shuffled by the same permutation", the US methods can perform well, and much better than the proposed algorithm, as long as A is full column rank (in this situation the underlying subspace is fully determined). To understand this, please read carefully the experiments of [Yao, 2021], and relevant papers listed below, (and also cited in [Yao, 2021]).
> >
> > The algorithm of (Zhang & Li 2020) is of statistical interest, and is not meant for practical usage. It is a rather weak baseline.
> >
> > To be serious, I recommend the authors to discuss all the papers listed below, each in 1-2 sentences, as a review of related works. One sentence is to describe what the proposed algorithm of the paper is, and another sentence is about the relation to the problem of interest. In fact, basically all papers that I listed below can be directly applied to the MUS problem here. And for the missing entries case, matrix completion + these algorithms would work, in different situations.
> >
> > A even more serious paper would take two representitive algorithms, one from the US literature and one from the MUS literature, and make careful experimental comparison. But I did not require this because I know what would happen. Instead, I believed the contribution of this paper is for the missing entries + missing correspondences case.
> >
> > Finally, the d-correspondence problem is a special case of the situation considered in [Yao, 2021], for the same reason that linear regerssion is a special case of US. This is about conceptual relation, is not about algorithm performance. On the other hand, in terms of algorithms, linear regression can be solved more efficiently than US, but it does not mean the d-correspondence problem can be solved more efficiently if collectively than solving the problem column by column, unless some paper proposes an algorithm and shows that is the case.
> >
> > I appreciate that the authors addressed my other concerns. But without proper treatment on related works, I do not agree the publication of the paper. Thus I updated my score to strong reject.
> >
> > I wish the area chair takes my comments into serious consideration.
> >
> >
> >  [REFERENCES]
> >
> > MUS:
> >
> > [1] https://arxiv.org/pdf/1704.07461.pdf
> >
> > [2] http://proceedings.mlr.press/v115/slawski20a.html
> >
> > [3] https://www.jmlr.org/papers/volume21/19-645/19-645.pdf
> >
> > US:
> >
> > [4] https://arxiv.org/abs/1810.05440
> >
> > [5] https://ieeexplore.ieee.org/document/9178410
> >
> > [6] http://proceedings.mlr.press/v97/tsakiris19a.html
> >
> > [7] https://arxiv.org/abs/1910.01623
> >
> > [8] https://arxiv.org/abs/2012.00123
> >
> > Subspace Learning:
> >
> > [9] https://arxiv.org/abs/1510.04390

---

> > > ### Author Response · Authors · 2021-11-24
> > > **Thank you for the important additional comments! We still hold our opinion about  the prior knowledge  on permutation and give references to stand up our point. We will update the literature review in one day in response to your suggestions.**
> > >
> > > We  thought that the statement of Reviewer CuTB "more aggressive noise model"  refers to the scenario with random missing values, and  there was no misunderstanding. We have edited the response to  Reviewer CuTB to make clear of this point.
> > >
> > > We politely disagree with the comments
> > > 1. "... US methods can perform well, and much better than the proposed algorithm, as long as A is full column rank ...".
> > > 2.  "... but it does not mean the d-correspondence problem can be solved more efficiently if collectively than solving the problem column by column ... "
> > >
> > > We still argue that  the prior knowledge can improve over US methods in the noisy scenario theoretically (which is common in real world). Consider the scenario: the subspace of  $A$ covers the that of $[A,B]$.  However, if A and B are corrupted by independent noise, the subspace of $[A,B]$ can not be exactly determined by $A$. In our opinion, leveraging the prior knowledge "some of the columns are shuffled by the same permutation" to resist the effect of noise is one of the main spirit for all the  MUS algorithms. There are many existing works for MUS problem, including those mentioned by the reviewer, supporting this claim. Here we provide an overview for these works:
> > > 1. A  theoretical discussion  is provided in the Section 4.2 of (Zhang & Li 2020 http://proceedings.mlr.press/v119/zhang20n/zhang20n.pdf ), which said that one can resist stronger noise if we know that more columns are shuffled by the same permutation.
> > > 2. Quoted from the page 5 of [1] https://arxiv.org/pdf/1704.07461.pdf:
> > > "As a result, we see that if $m \gg \log n$, then the permutation does not play much of a role in the
> > > problem, and the rates resemble those of standard linear regression. Such a general behaviour
> > > is expected, since a large $m$ means that we get multiple observations with the same unknown
> > > permutation, and this should allow us to estimate $\Pi$ better. "
> > > 3. Quoted from the conclusion of [2] http://proceedings.mlr.press/v115/slawski20a.html: "A large number of (not too strongly
> > > correlated) outcomes appears to be crucial to the success
> > > of the approach: it not only allows for a clear separation
> > > between correctly matched data and mismatched data,
> > > but also significantly facilitates permutation recovery as
> > > elaborated in §3."
> > > 4. Quoted from the conclusion of [3] https://www.jmlr.org/papers/volume21/19-645/19-645.pdf "A key result in this paper asserts that the availability of multiple,
> > > linearly independent response variables (as measured by the stable rank of the regression
> > > coefficients) considerably simplifies the problem as it increases separability"
> > >
> > > All the works mentioned above show the same thing: We can do better if we know that more columns are shuffled by the same permutation. However, this claim is only made theoretically not practically.
> > >
> > > We agree that the algorithm of (Zhang & Li 2020) may not be a  state-of-the-art algorithm for the MUS problem, and one of the main reasons we choose it is that it is replicable in experiment, while the other works and [Yao,2021] did not release their implementation and we failed to replicate them. Even though, we think that this algorithm is enough to convey an important message that the idea "First do the matrix completion and then run the MUS algorithm " does not perform  well. Because the first step "matrix completion with unknown correspondence" would bring in too much error  and could affect the behaviour of MUS algorithm in the second step significantly.
> > >
> > > In response to the reviewer's suggestion, we will update our literature review in one day to cover the following points:
> > >
> > > 1. Discussion for the difference between MUS problem and US problem. The main point is to show that, for a general noisy problem, (Zhang & Li 2020)[1][2][3] have proved theoretically that we can utilize the prior knowledge "some of the columns are shuffled by the same permutation" to improve over US methods. However, some of US algorithms like the these adopted by [Yao,2021] can be more suitable for the dense permutation scenario in practice.
> > > 2. Brief review for the methodology of all the listed  papers. We will also state explicitly that the algorithm of (Zhang & Li 2020) is not state-of-the-art and the main point we use it as baseline is to show that "direct matrix completion+MUS algorithm" does not perform well.
> > > 3. More discussions for the paper [Yao,2021]. Firstly, they adopted  US methods for the dense permutations scenario, which do not exploit the prior knowledge on permutation. Secondly, they  adopted the robust regression framework for the sparse permutation scenario, but the sparse permutation scenario is not our target scenario.
> > >
> > > If you still have concerns that are not addressed by the response above, please do let us know and we are pleased to give a follow-up response. Thank you again for all these important comments!
> > >
> > > We also wish the area chair can take a serious consideration for our response, thanks!

---

> > > > ### Comment · Reviewer_2Tsx · 2021-11-24
> > > > **Thanks for your reply**
> > > >
> > > > 1. The authors quoted [1-3] and (Zhang & Li 2020), to claim that the more (not too strongly correlated) columns A or B has, the problem is easier to solve. This is exactly because in that situation the underlying subspace is much easier to identify individually from A or B. Of course, the authors made an important point here, and this should be included in the paper, along with references. Finally, the authors disagreed my point 2 above, but please note that I have written "unless some paper proposes an algorithm and shows that is the case". Finally, it is also important to point out that the algorithm of [1] is not robust to noise, and the algorithms of [2] and [3] are for sparse permutations.
> > > >
> > > > 2. The authors claim that high noise might lead to inaccurate subspace computation. But finding the subspace from A or B is just a task of PCA, and PCA is highly robust to Gaussian noise (it is just not robust to outliers).
> > > >
> > > > 3.  About replicating prior algorithms: The algorithms of [1-3] are easy to implement. The former involves a SVD computation and sorting, the latter two can be implemented using standard CVX package. For the code of [Yao 2021], check out: https://github.com/yaoyzh/Unlabeled_PCA_NeurIPS2021  (of course this is not required for comparison as it is released only recently, and the paper here seems to be an independent effort than [Yao 2021]). For other US methods, codes might be found on the webpage of corresponding authors. For example, check out: https://sites.google.com/site/manolisctsakiris and https://sites.google.com/site/slawskimartin/about
> > > >
> > > > 4. The authors disagreed with my claim:
> > > > "... US methods can perform well, and much better than the proposed algorithm, as long as A is full column rank ...".
> > > > Their arguments for disagreeing with it are not convicing, because there are no experiments to show that, and because the references are not contradictory to my claim (as explained above). On the other hand, I made a quick attempt at running matrix completion + US methods using the submitted code to prove I am correct. But I failed to run them, because the submitted code has no comments, and matrix completion is integrated with permutation recovery in a single function (yet that is a feature of the proposed algorithm). I wish the authors have sufficient interest in validating my claim by experiments (using the links to the code above). Of course, such experiment does not have to appear in the paper, but the important thing is to know which methods work, under which conditions, and discuss them thoroughly in related works.
> > > >
> > > > 5. Finally, please note that in the experimental setting (synthetic), the rank is 5, and A and B are Gaussian and have 100 rows and more than 20 columns. It is reasonable to guess that, directly apply matrix completion approaches, individually to A and B, might complete their entries quite good (provided that not too much entries are missing). Hence, please make sure to verify, via experiments,  matrix completion would make large errors when completing A (resp. B). The point here is to complete them individually, not collectively. Otherwise the error can be large.
> > > >
> > > > Let me summarize my point of view on the paper:
> > > >
> > > > Strength: the paper made some interesting theoretical discovery about the problem of matrix recovery with unknown correspondences (MRUC), and proposed an algorithm to solve MRUC with missing entries by minimizing a single objective function (in contrast to a straightforward approach which does matrix completion + (multivariate) unlabeled sensing).
> > > >
> > > > Weakness: Insufficient discussion on related works. This caused sereve misunderstanding to readers (reviewers) who are either familar or unfamiliar with the literature. But this might be eventually fixed, as when I pointed out them seriously, the authors took actions promptly. I wish the authors also take a look at any other possible related papers (via google search) in the US or MUS literature. I might miss something here perhaps.
> > > >
> > > > I keep my score there, and I have finished my review on the paper. I wish other reviewers and the area chair take a look at the discussion here, and make a fair decision. Thanks.

---

> > > > > ### Author Response · Authors · 2021-11-25
> > > > > **We feel very sorry that our previous  literature review has bother you so much. We will conduct more new experiments in two days as you suggested, and hope that you could give us one more opportunity make our paper more correct and informative.**
> > > > >
> > > > > Dear reviewer 2Tsx,
> > > > >
> > > > > We feel very sorry that our previous inadequacy in literature review on some of related works has bother you so much. But we still want to thank for your important comments on our paper, and it indeed helped us a lot on improving the quality of literature review section.
> > > > >
> > > > > Now, we will try our best to make every sentences truthful and justifiable by conducting a series of new experiments and reading more related papers seriously in two days. We hope that you could still give us one more opportunity make our paper more correct and informative.
> > > > >
> > > > > We believe that the main conflict point between us is our understanding towards "whether MUS is better than US". At first, we indeed believe that "MUS is better than US", as we had learned from many theoretical result in existing works. But thanks to the literature listed by the reviewer, we now learn that, from the practical angle, it still remains unclear whether MUS algorithms can be better than US algorithms in existing literature.
> > > > >
> > > > > Therefore, as this is an unknown and interesting question in this filed, we will conduct experiments to investigate the performance of MUS algorithms and US algorithms comprehensively on several different scenarios. It could be expected that we can learn from the experiment results that under what scenario MUS algorithms are better, and under what scenario US algorithms are better. Our experiments will mainly based on the codes pointed out by the reviewer.
> > > > >
> > > > > We will also modify our literature review extensively to show the the following facts pointed out by the reviewer:
> > > > >
> > > > > 1.Discussion for MUS problem and US problem. To show that existing literature only prove that "MUS is better than US" theoretically, not practically.
> > > > >
> > > > > 2.Brief review for the methodology of all the listed papers. We would like to categorize them into three classes for discussion of their suitable scenario: algebraic geometry based methods, relaxation based method, and robust regression/subspace learning based method.
> > > > >
> > > > > 3. More discussions for the paper [Yao,2021]. We will admit that their method designed for the dense permutation scenario could behave properly combining with matrix completion for our MRUC problem.
> > > > >
> > > > > All in all, we still hope that the reviewer could kindly reconsider his/her evaluation after seeing our new experiment results and literature reviews, because we believe that our discussions arose in the rebuttal period will benefit the other people in the filed of US and MUS, while our previous inadequacy in literature review does not degrade our main contribution in this work too much.
> > > > >
> > > > > Again, we appreciate for all the efforts you made for improving our paper! We are looking forward to your final response! Thank you!

---

> > > > > > ### Comment · Reviewer_2Tsx · 2021-11-25
> > > > > > **Thanks for your reply**
> > > > > >
> > > > > > I will reconsider my score after that. Thanks.

---

> > > > > ### Author Response · Authors · 2021-11-25
> > > > > **We have finished a thorough experiment to discuss the relationships among MRUC , US and MUS as you suggested, and have reached some interesting conclusions.**
> > > > >
> > > > > Updated: Since we cannot update the supplementary materials to include the codes for this experiment in this period. If the reviewer is still interested in seeing the codes, we will post them in the comment block.
> > > > >
> > > > > The main concern of the reviewer that need to be addressed is:
> > > > >
> > > > > When do US algorithms perform better than MUS algorithms and our MRUC algorithms?
> > > > >
> > > > > To answer this question, we conduct experiments using three algorithms on two different scenarios.
> > > > >
> > > > > We use the algorithms:
> > > > >
> > > > > * MRUC: Our proposed algorithm $M^3O$.
> > > > > * US: CCV-min algorithm used in [Yao,2021], because it achieves the best performance in the paper. We use the codes in https://github.com/liangzu/CCVMIN.
> > > > > * MUS: The algorithm in [Han, 2020]. We failed to find codes of Slawski's papers, and LevSort is not suitable for noisy scenario.
> > > > >
> > > > > In the following experiments, we consider the scenario without missing values.
> > > > >
> > > > > The first experiment is done on the following low-rank scenario:
> > > > >
> > > > > * Low-rank random matrix: $M=[A,B]=$randn(100,5)\*randn(5,100)+0.1\*randn(100,100) , where the columns number of $A$ is 40. The permuted matrix is
> > > > > $M_o=[A,PB].$
> > > > >
> > > > > In this experiment, we also propose improved versions of US algorithm and MUS algorithm, by replacing their inputs $A$ and $PB$ with their top five left singular vectors
> > > > > $U_A$ and $U_{PB}$. This process can be viewed as a simple version of the first step subspace learning in [Yao,2021]. For the US algorithm, we randomly choose one column of $PB$ as the response variable.
> > > > >
> > > > > Here we provide the result by varying the sparsity of $P$, the numbers in table are permutation error
> > > > > $d_H(\hat \pi_P,\pi_P)$.
> > > > >
> > > > > |Algorithms\Sparsity|10|	30|	50|	70|	90|
> > > > > | :---        |    :----:   |          ---: |  ---: |  ---: |  ---: |
> > > > > |$M^3O$|	0|	0|	0|	0|	0|
> > > > > |CCVmin|	100|	96|	99|	100|	98|
> > > > > |CCVmin\_SVD|	62|	83|	69|	70|	66|
> > > > > |MUS|	|0	|0|	0|	37|	96|
> > > > > |MUS\_SVD|	0|	0|	0|	10|	98|
> > > > >
> > > > > An interesting fact is that, although CCV-min performs bad in permutation recovery, it performs very well in vector recovery (We remark this phenomenon is not a contradiction to the results in [Yao, 2021], as they only reported the results for vector recovery.). The numbers in following table are the relative error
> > > > > $||\hat B_i-B_i||_F^2/||B_i||_F^2$.
> > > > >
> > > > > |Algorithms\Sparsity|10|	30|	50|	70|	90|
> > > > > | :---        |    :----:   |          ---: |  ---: |  ---: |  ---: |
> > > > > |CCVmin|	6e-3|4e-3	|4e-3|3e-3|4e-3|
> > > > > |CCVmin\_SVD|8e-4|2e-3|6e-4|1e-3|1e-3|
> > > > >
> > > > > We suspect that this is because the univariate linear system could not guarantee a unique solution for permutation matrix. On the contrary, MUS does better in permutation recovery. Combining with existing theoretical results, we can naturally propose a hypothesis: the prior knowledge "some of the columns are shuffled by the same permutation" can improve the permutation recovery, not the vector recovery. Of course, a serious study of this hypothesis is beyond the scope of our paper.
> > > > >
> > > > > Finally, it is intuitive that our algorithm perform consistently the best, because it exploits the low-rank structure of data matrix more directly.
> > > > >
> > > > > Then we did another experiment on the following regression scenario:
> > > > >
> > > > > Regression-form matrix: First generate $A=$randn(50,5)\*10, $x=$randn(5,45)/10, $y=Ax+$0.05\*randn(50,50), then generate $M=[A,y]$ and $M_o=[A,Py]$.
> > > > > Here we also provide the permutation recovery result by varying the sparsity of $P$. The numbers in table are permutation error.
> > > > >
> > > > > |Algorithms\Sparsity|5|	15|	25|	35|	45|
> > > > > | :---        |    :----:   |          ---: |  ---: |  ---: |  ---: |
> > > > > |M3O|	49|	50|	49	|48|	50|
> > > > > |CCVmin|	13|	13|	8	|7|	12|
> > > > > |MUS|	0	|0	|0	|0|	45|
> > > > >
> > > > > It is natural to see that our algorithm perform worse than the other two algorithms in this scenario, because now $A$ and $y$ have significantly different magnitude. This is consistent with our theoretical discussions in remark 1.
> > > > >
> > > > > Conclusions:
> > > > >
> > > > > * Our algorithm is more suitable with data that has a low-rank structure and is partitioned evenly, whereas MUS and US may be more suitable with data that has a regression form.
> > > > >
> > > > > * The prior knowledge "some of the columns are shuffled by the same permutation" may only improve the permutation recovery, not the vector recovery.
> > > > >
> > > > > * An additional comments for [Yao,2021] is that, in the synthetic data experiment, they did not report the result of permutation recovery and only report the result of vector recovery. On the contrary, our paper demonstrate explicitly for the result of permutation recovery.
> > > > >
> > > > > * For the scenario low-rank matrix with missing values, it is naturally to infer that the performance gap between our method and the US/MUS method could enlarge. Of course, we would conduct this experiment in no time if the reviewer is still interested in seeing a concrete result.
> > > > >
> > > > > Additional remark: For all our experiments in the paper, the matrix completion was indeed done independently as the reviewer said.
> > > > >
> > > > > We are pleased to do any further experiments if you are still interested in extra related results. Thank you very much!

---

> > > > > > ### Comment · Reviewer_2Tsx · 2021-11-25
> > > > > > **Thanks for your experiments**
> > > > > >
> > > > > > It is another interesting contribution to compare US/MUS/MRUC methods.
> > > > > >
> > > > > > "we can naturally propose a hypothesis: the prior knowledge "some of the columns are shuffled by the same permutation" can improve the permutation recovery, not the vector recovery."
> > > > > >
> > > > > > Incorrect. If you could recover the permutation (vector) very well, then, in general you could recover the other unknown also very well. If the vector is known, recoverying the permutation  reduces to a linear assignment problem in MRUC or MUS, and a sorting problem in US. The reason CCVMIn did not recover the permutation correctly was explained theoretically in (Pananjady, TIT 2018): unique/approximate permutation recovery requires very high SNR; see also (Hsu, NIPS 2018) for similar results on vector recovery in US. Here you set a high noise 0.1. This is in contrast to the MUS case, where more columns in B allow for more accurate permutation recovery under more noise. Based on this, with the estimate of the vector from CCVMin, and with the prior knowledge that B has undergone the same permutation, one could use linear assignment for better permutation recovery. Similarly, since M^3O did a good job in permutation recovery, a refinement step of linear regression would improve vector recovery (but it might not, as M^3O already used such prior knowledge, while CCVMin did not use it during the algorithm).
> > > > > >
> > > > > > "Finally, it is intuitive that our algorithm perform consistently the best, because ..."
> > > > > >
> > > > > > Please do not say that (e.g., it might not be intuitive to somebody else, like me), unless you showed that M^3O has even smaller errors for vector recovery than all available algorithms (as different algorithms have different working regimes), and unless M^3O could run faster. In general, write humbly to avoid suspection. Thanks.
> > > > > >
> > > > > > Finally, please be familiar with the results of (Pananjady, TIT 2018) and (Hsu, NIPS 2017) that I mentioned above (if you have not). Make a solid review on related works; some of our discussions could be put into the paper in a more organized way.
> > > > > >
> > > > > > Thanks for the efforts, I restored my socre. I believe I have made sufficient efforts here to reviewing the paper, and I would no longer reply. Thanks for your understanding.

---

> > > > > > > ### Author Response · Authors · 2021-11-25
> > > > > > > **Many thanks for all the efforts you have made to improve this paper!**
> > > > > > >
> > > > > > > Dear Reviewer 2Tsx,
> > > > > > >
> > > > > > > Thank you very much for providing us so many critical and constructive advice.
> > > > > > >
> > > > > > > We will take all our discussions and your suggestions into serious consideration for the revised paper!
> > > > > > >
> > > > > > > Wish you all the best!

---

> > > ### Author Response · Authors · 2021-11-24
> > > **More comments for the relationship between US algorithms and MUS algorithms.**
> > >
> > > We would like to discuss more about the relationship between US algorithms and the MUS algorithms.
> > >
> > > It is easy to see that all the MUS algorithms can be directly extended to the US problem, as US problem is just a special case of MUS problem.
> > >
> > > However, not all the existing US algorithms can be directly extended to a MUS algorithms (Of course, a naive extension is to apply US algorithms column by column).
> > >
> > > To our knowledge, existing works for MUS problem like (Zhang & Li 2020)[1][2][3] all demonstrate that MUS algorithms will perform better than their US extension (to apply the MUS algorithms column by column) theoretically and empirically, and some of the works (like [2]) even show that their MUS algorithms can be better than the other existing US algorithms even for the US problem.
> > >
> > > Despite of this, in practice, all the MUS algorithms in (Zhang & Li 2020)[2][3] can not deal with dense permutation, whereas the US algorithm adopted by [Yao, 2021] can indeed deal with these scenarios.
> > >
> > > Therefore, we agree with reviewer 2Tsx that it is still reasonable for us to consider adopting the naive extension of existing US algorithm as baselines for our MRUC problem, and will modify our previous opinion in literature review.

---

### Official Review · Reviewer_CuTB · 2021-10-28

**Correctness:** 4
**Technical Novelty And Significance:** 3
**Empirical Novelty And Significance:** 3
**Recommendation:** 8
**Confidence:** 3

**Main Review:**

Strengths:

1. I think this is an interesting problem and the paper is an interesting second step that comes up with an algorithm that more closely respects the structure of the problem than prior work.


**Summary Of The Paper:**

This paper studies the following matrix recovery problem, given an observation matrix $M_o = [A, \tilde P B]$, where $\tilde P$ is an unknown permutation matrix, recover the underlying matrix $M = [A, B]$. The underlying assumption is that the original matrix $M$ is low-rank. They study this in the settings where there are missing entries as well as in the presence of bounded additive Gaussian noise.

Prior work has more extensively considered the version of the problem of multivariate linear regression with unknown correspondence, where the goal is to solve $\min_{P, W} \|Y - PXW\|^2_F$ where $W$ are the regression coefficients, $X$ are the permuted input, and $Y$ is the output.

[1] considers the problem that is solved in this paper as well, but use an extension of the algorithm meant for the regression version of the problem which seems to suffer from the inability to extend to more aggressive noise models.

Their proof relies on studying how the permutation $P$ changes the rank of the matrix $M$, the authors demonstrate that properties of the cycle decomposition of the permutation $P$ control the rank, qualitatively, the effect of $P$ on $\text{rank}(M)$ becomes stronger the more rows it permutes and contains less cycles. The program they analyze is the standard nuclear-norm relaxation for the rank function.

They show that if (1) the singular values of $A, B$ are comparable, (2) the column-space of $M$ can be approximated by the column space of one of its submatrices and (3) The columns of $M$ do not contain any duplicated elements; then the difference between the recovered permutation and original permutation is small (as long as the rank of $M$ is small, the additive noise is bounded.

To extend this to the case of missing entries requires some technical work which changes the program to nuclear-norm regularized, frobenius norm regression on the coordinates that are present. They solve the problem via an alternating optimization approach -- the regression problem is convex in the unknown matrix $\widehat M$, and they interpret the problem of recovering the permutation $P$ as an "entropic optimal transport" problem, which is strongly convex.

The authors also run a fairly detailed set of experiments that demonstrate the performance of their algorithm.


[1] Unlabeled principal component analysis. -- Yunzhen Yao, Liangzu Peng, and Manolis C Tsakiris'21

**Summary Of The Review:**

I think the results are technically interesting and the paper has good motivation, I vote to accept this paper.

---

> ### Author Response · Authors · 2021-11-15
> **Thank you for your compliment!**
>
> Thank you for your kindly support and recognizing our contribitions!
>
> As mentioned by Reviewer 2Tsx, we politely point out that it may be more precise to state that "[1] considers the problem that is solved in this paper as well, but use an extension of the algorithm meant for the regression version of the problem which seems to suffer from the inability to extend to the models with missing values."
>
> Any additional comments are welcome!

---

### Official Review · Reviewer_YsMp · 2021-11-01

**Correctness:** 4
**Technical Novelty And Significance:** 4
**Empirical Novelty And Significance:** 4
**Recommendation:** 10
**Confidence:** 3

**Main Review:**

The strengths are clear - good theoretical analysis, supported with novel practical algorithms.
My only criticism (and a minor one) is that the image face example seems contrived.
It would be good to see more real world applications (and ones not contrived).

**Summary Of The Paper:**

The paper take a scenario - such as recommender systems *federated from different sources so one does not now correspondences* of low rank matrix recovery from partially observed (missing data) and slightly corrupted (small noise) data. It then shows that recovery os theoretically possible and also that an obvious current algorithm candidate will fail but offers a substitute algorithm that works well on the examples given.

**Summary Of The Review:**

The application to situations where one knows there are some correspondences (but not the correspondences themselves, for privacy reasons) is sufficiently of general interest. The analysis seems correct and an important contribution. This is further supported with a novel algorithm that is reasonably sufficiently demonstrated on synthetic and real data.
In general it seems close to a model paper.

---

> ### Author Response · Authors · 2021-11-15
> **Thank you for your appreciation! We believe that  the MRUC problem could arise in more scenarios in the future, and our work can serve as a general framework to deal with the missing correspondence issue.**
>
> Thank you for your appreciation!
>
> We do agree with you that the face recovery problem might not be very practical. But this experiment indeed demonstrate a very important property of algorithm: Our algorithm indeed exploit the low-rank structure in data very well.
>
> We believe that the MRUC problem could arise in more scenarios, since low-rank property is widely observed in data from many applications. For example, as mentioned in the conclusion section, the gnome assembly problem [Huang and Madan, 1999], the video pose tracking problem [Ganapathi et al., 2012] and the privacy-aware sensor networks [Gruteser et al., 2003], etc. We believe that our work can serve as a general framework to deal with the missing correspondence issue in these scenarios. Due to limited time, we cannot manage carry out experiments for all these potential applications.
>
> Any additional comments are welcome!

---

### Official Review · Reviewer_Wx9p · 2021-11-02

**Correctness:** 3
**Technical Novelty And Significance:** 3
**Empirical Novelty And Significance:** 3
**Recommendation:** 5
**Confidence:** 3

**Main Review:**

This paper is well-written and easy to follow. I like the writing style of the paper that uses bold first sentences of important paragraph to summary the main point.
The result of the paper is interesting, and could be important as it is one of the few papers on this subject.
I like the cute result analyzing the perturbation of the rank of the matrix after permutation.

Some concerns:
- There is the parameter $\epsilon_3$ in Assumption 3, which appears in the upper bound as the factor $1/\epsilon_3^2$. Q1: What is the expected scale/order of $\epsilon_3$ to make the upper bound non-trivial? Q2: Does that order make sense? If I randomly sample a $n$-dim unit vector and calculate the minimal distance between its entries, then I would guess it could be at least smaller than the order of $n^{-3/2}$, because the order of each entry is $O(n^{-1/2})$, and there are $n$ entries. If I set $\epsilon_3 = O(n^{-3/2})$ by that argument, it becomes a scale of $n^3$ in the upper bound, which might make the upper bound greater than $n$ and thus trivial (since the distance between two permutations is at most $n$).

- Assumption 1 and Assumption 2 basically say that, the singular values and left singular vectors of A and B are close. With these assumptions, one naive algorithm in mind would be, to do SVD of $A$ and the permuted matrix $B$, and find a best matching between their rows, to recover the permutation. For example, one can sort the singular values of each matrices to find a matching between singular values, and use the left singular vectors to do some adjustment for very close singular values (because say if A has a duplicated singular values, we don't know how to match them to the singular values of the permuted B since we don't know which order to map. In this case we can use left singular vectors to help). How do the author think this algorithm would perform, and is there any intuitions that the proposed algorithm in the paper should outperform this method?

Some typos:
- page 3, 'with multiple permutation' -> 'with multiple permutations'
- page 6, 'the number of possible correspondence increase' -> 'the number of possible correspondence increases'
- Page 9, 'provides a general aolution' -> 'provides a general solution'

---------------------- Score after revision -----------------------

I keep my original score, due to the concern that, one might need strong assumptions to make the upper bound of Theorem 1 non-trivial: the parameter $\epsilon_3$ typically contributes O(n^3) in the upper bound. To make the upper bound non-trivial (that is, o(n)), one might need strong assumptions on other parameters $\epsilon_1, \epsilon_2, \sigma$. Currently I'm not sure if those assumptions make sense in reality.



**Summary Of The Paper:**

This paper studies the matrix recovery with unknown correspondence problem, where we have two data matrices from different sources, with unknown correspondence, and the task is to estimate the underlying correspondence. Mathematically, one can imagine that the row of one data matrix is randomly permuted, and the goal is to recover the permutation. Of course this problem cannot be solved without any conditions, and the condition the author consider is that the joint matrix of the data is low rank.

Contribution of the paper:
- The authors provide theorems to analyze how the random permutation perturb the rank of the data matrix.
- The authors propose to solve this problem by a nuclear norm minimization problem, and provide theoretical upper bound on the distance between their estimate and the true underlying permutation.
- The proposed algorithm can work with multiple unknown correspondence, dense permutations, and missing values, which are not achieved by another paper (Yao et al., 2021) on the same subject.

**Summary Of The Review:**

Overall I like the idea of the paper and the presentation, however I have one concern about the Assumption 3 and the possibility that the main result (upper bound) in the paper might be trivial in some cases, and hope the author could address that in the rebuttal period.

---

> ### Comment · Reviewer_2Tsx · 2021-11-02
> **about one concern in the review of Reviewer Wx9p**
>
> Dear Reviewer Wx9p,
>
> Thanks for your careful review. Your concern about Assumptions 1 and 2 and your following thought remind me of paper [1], where an algorithm that is similar to what you described was presented. This is an algorithm for multivariate unlabeled sensing, and is called LevSort. It provably solves (1) in the noiseless case, provided the column spaces of A and B are the same, and it might be able to tolerate slight noise. This, again, implies that the paper has an insufficient discussion on related works.
>
> @authors, please address this important concern. More generally, a very straightforward algorithm is to combine matrix completion + (multivariate) unlabeled sensing algorithms. The aurhors should discuss this important aspect, and make some comparisons, experimentally or conceptually.
>
> That said, I believe the main contribution of the paper is an algorithm for the case of missing entries, as well as some interesting theory.
>
> [1] https://arxiv.org/pdf/1704.07461.pdf
>
>
> Update (Nov. 23, 2021): I made this comment after Reviewer Wx9p released his/her review, but I did not set the readership correctly.

---

> > ### Author Response · Authors · 2021-11-24
> > **Thank you for pointing out this important reference!**
> >
> > Thank you for pointing out this important reference! We will update our literature review in one day to cover this paper, as stated in the responses for reviewer 2Tsx.
> >
> > As we have discussed in the responses for reviewer 2Tsx, in our paper, we adopt a typical MUS algorithm proposed by (Zhang & Li 2020) to convey an important message that the idea "First do the matrix completion and then run the MUS algorithm " does not perform well. Of course, it is also possible to replace the algorithm of (Zhang & Li 2020) with other state-of-the-art algorithm. But we believe the conclusion will remain the same, because, conceptually, the first step "matrix completion" would introduce too much error with the unknown correspondence, and could affect the behaviour of the second step significantly.

---

> ### Author Response · Authors · 2021-11-15
> **Thank you for all these important comments!  We have proofread our paper carefully and added an additional section in appendix to discuss more about the asymptotic behavior of Theorem 1.**
>
> Thank you for pointing out these typos and we have proofread our paper carefully and made revisions.
>
> As pointed out by reviewer 2Tsx, we will correct our statement in this paper and  mention that it is possible (Yao et al., 2021) can deal with multiple unknown correspondence, dense permutations on the same subject, and our main contribution will be lie on dealing with missing values.
>
> **Q1. When is the upper bound in (10) nontrivial?**
>
>  Thank you for raising this interesting and important question to us! We have added an additional section A.2 in the appendix to discuss more about the asymptotic behavior of Theorem 1.
>
> Indeed, with $\epsilon_1,\epsilon_2,\sigma$ fixed, if the elements of the matrix follow uni(0,1) i.i.d, then the r.h.s of  inequality (10) is at least O($n^3$) w.h.p. However, we remark that the  asymptotic behavior ($n\to\infty$) of Theorem 1 is more related to $\epsilon_1,\epsilon_2,\sigma$ instead of $\epsilon_3$.
>
> We claim that: For any $n$, as long as $\epsilon_3>0$,  we can have a nontrivial upper bound provided that $\epsilon_1,\epsilon_2,\sigma$ sufficiently small.
>
>  In fact, without $\epsilon_1\to 0,\epsilon_2\to 0,\sigma\to 0$, the original matrix will be impossible to recover by minimizing nuclear norm for sufficient large $n$. In this way, once the upper bound for permutation error becomes trivial, we can say that the ground-truth permutation  is impossible to recover.
>
> Here we provide a simple example to validate our claim: Suppose that the original matrix is $M=[u,u]+W$, where the elements of $W$ follow $\mathcal{N}(0,\sigma^2)$ and $u\in\mathbb{R}^n$ is a random vector whose elements follow uni(0,1) i.i.d. We can construct a permutation matrix $P$ with $H(\pi_P)=n$, such that the following bound holds w.h.p,
> $$
> \bigg|||[u,Pu]||\_* -||[u,u]||_*\bigg|\leq ||Pu-u||_2= O(n^{-\frac{1}{2}}\log(n)).
> $$
> The above result is based on David and Nagaraja's book "Order Statistics"; see page 135.
>
> On the other hand, since  $||W||\_* \approx O(\sigma n)$ w.h.p, if we need  $||[u,Pu]+W||\_\*>||[u,u]+W||_*$, we at least require that $\sigma=o(n^{-\frac{3}{2}}\log(n))$. Otherwise, it will be impossible to distinguish the matrices $[u,Pu]+W$ and $[u,u]+W$ through the value of nuclear norm.
>
> Finally, for this simple example, since $\epsilon_1=\epsilon_2=0$ and $\epsilon_3$ is at most $O(n^{-3/2})$ w.h.p, we can show that the r.h.s of (10) is at least $O(n^{\frac{5}{2}}\sigma^{\frac{1}{2}})$ w.h.p. Therefore, we at least require that $\sigma=o(n^{-5})$ to guarantee a constant upper bound.
>
> This simple example well shows that  the  asymptotic behavior ($n\to\infty$) of Theorem 1 is more related to the other parameters instead of $\epsilon_3$.
>
>
> **Q2. A naive baseline.**
>
>  Thank you for pointing our this insight and this interesting algorithm!
>
> In our opinion, the main drawback of this algorithm  is that it can neither be adapted to deal with missing values nor handle the case where there are multiple correspondence. Hence it may have limited practical value.
>
> Besides, we believe that this algorithm is somewhat equivalent to the MUS algorithm mentioned in the introduction. This is because it can be mostly described as the unlabeled sensing problem for right singular vectors: Suppose that the $U$ and $U'$ are the left singular vectors of $A$ and $B$, it aims to solve
> $$
> \min_{P} ||U-PU'||_F^2.
> $$
> Therefore, we believe that  the performance of this algorithm can be inferred from our discussion in the introduction section and the numerical experiments.

---

### Author Response · Authors · 2021-11-15
**Summarization for the modifications made to our paper.**

* We have proofread our paper carefully and polished the language for the revised version.
* We have extended the reply to the Q1 of Reviewer Wx9p and added it as A.2 in the appendix so as to provide  more information about the asymptotic behavior of Theorem 1.
* We have added the discussion of the limit and future direction for Theorem 1 in the conclusion section
* We have add a new sentence to mention that problem (16) is strongly-convex w.r.t 1-norm, with corresponding reference.
* We have modified the review for [Yao, 2021] in the introduction section as follows:
  * Remove the statements that  [Yao, 2021] can not deal with multiple correspondence and dense permutation.
* We have added two more experiment results in A.9 of the appendix:
  * The discussion for the hyperparameter $\lambda$ (the penalty coefficient for nuclear norm).
  * The performance by varying the distance between the initialization and the optimal solution.
* We have provided a more serious literature review for more related works, especially the references mentioned by the reviewer 2Tsx in the revised version. (Due to page limit, we placed this content into appendix.)
* We have added the extra experiments in our response to reviewer 2Tsx into appendix A.10, together with some important implications of these experiments.

---

### Comment · Area_Chair_EaNJ · 2021-11-24
**reaching a conclusion on this paper**

Hi everyone, I hope you've had a chance to read each other's reviews and the author responses to each review. There are some serious discrepancies in the scores in this paper (a 10 and a 1!), so please take a look, and let me know what you think, thanks.

---

### Decision · Program_Chairs · 2022-01-20

**Decision:**

Reject

**Comment:**

Overall this paper was discussed at length given the high variance in scores, and it was ultimately felt that the paper was a borderline paper and there was not enough enthusiasm to warrant acceptance. Several concerns in the discussion could not be resolved, in particular the bounds might not be tight, or even useful, and more explanation on the dependence of various parameters involved and assumptions involved is needed. Specifically, as pointed out by a reviewer, there is a concern about the parameter epsilon_3. It seems for natural input distributions epsilon_3 would be so small that the upper bound would scale as n^3 (given the 1/epsilon_3^2 dependence), which is then trivial since it is larger than n. The reviewers were not satisfied with the authors response regarding this.